evolution

sperm storage, spermatheca, sperm competition, sperm, reproduction

**Author for correspondence:**
Ben R. Hopkins
e-mail: brhopkins92@gmail.com

# Structural variation in *Drosophila melanogaster* spermathecal ducts and its association with sperm competition dynamics

Ben R. Hopkins[1,2], Irem Sepil[1] and Stuart Wigby[1,3]

[1]Department of Zoology, University of Oxford, Oxford OX1 3SZ, UK
[2]Department of Evolution and Ecology, University of California – Davis, One Shields Ave., Davis, CA 95616, USA
[3]Faculty Biology, Applied Zoology, Technische Universität Dresden, 01069 Dresden, Germany

BRH, 0000-0002-9760-6185; IS, 0000-0002-3228-5480;
SW, 0000-0002-2260-2948

The ability of female insects to retain and use sperm for days, months, or even years after mating requires specialized storage organs in the reproductive tract. In most orders, these organs include a pair of sclerotized capsules known as spermathecae. Here, we report that some *Drosophila melanogaster* females exhibit previously uncharacterized structures within the distal portion of the muscular duct that links a spermatheca to the uterus. We find that these 'spermathecal duct presences' (SDPs) may form in either or both ducts and can extend from the duct into the sperm-storing capsule itself. We further find that the incidence of SDPs varies significantly between genotypes, but does not change significantly with the age or mating status of females, the latter indicating that SDPs are not composed of or stimulated by sperm or male seminal proteins. We show that SDPs affect neither the number of first male sperm held in a spermatheca nor the number of offspring produced after a single mating. However, we find evidence that SDPs are associated with a lack of second male sperm in the spermathecae after females remate. This raises the possibility that SDPs provide a mechanism for variation in sperm competition outcome among females.

## 1. Introduction

Female insects commonly store sperm for extended periods after mating [1,2]. Where those females mate multiply, sperm from rival males may be stored simultaneously and have to compete

over access to limited fertilizations—a process known as sperm competition [3,4]. How the physiology of sperm storage influences the outcome of sperm competition remains a major question in the field of evolutionary reproductive biology [5,6].

Maintaining the long-term viability of sperm presents challenges [7]. Retaining sperm within specialized storage organs is thought to help buffer against the risk of desiccation, thermal stress, immune attack and the mutagenic action of oxidative stress. In most insects, the storage organs are sclerotized capsules known as spermathecae, the number and morphology of which varies between species [8]. These organs show clear adaptations to long-term sperm use including tight control of sperm release [9] and the production of viability-enhancing secretions [10,11]. Consequently, variation in the physiology of sperm storage organs is likely to have correlated effects on female reproductive performance.

In *Drosophila melanogaster*, females store the majority of received sperm in the seminal receptacle, a novel tubular structure found only in certain acalyptrate Dipteran families [8,12]. Variation in seminal receptacle morphology has known consequences for sperm competition outcome: longer seminal receptacles benefit longer sperm in both displacing rival sperm from storage and themselves resisting displacement [13,14]. The remaining sperm are stored in two (or rarely three) spermathecae, which consist of chitinized capsules that connect to the uterus via a muscular duct [15,16]. As in the seminal receptacle, sperm stored within the spermathecae can be displaced by an incoming ejaculate [5], but variation in the morphology of *D. melanogaster* spermathecae remains largely uncharacterized. However, there is evidence of between-population divergence in spermathecal shape in *D. affinis*, a member of the *Sophophora* subgenus to which *D. melanogaster* belongs [17]. But whether this variation has consequences for sperm storage patterns or sperm competition outcome remains untested.

Here, we report the identification of novel structures in the spermathecal ducts of a subset of *D. melanogaster* females. We test for differences in the incidence of these 'spermathecal duct presences' (SDPs) between female genotypes, age classes, and between mated and virgin females. We then test the hypotheses that SDPs are associated with compromised sperm release, storage, and offspring production, features that would implicate them in determining sperm competition outcome.

# 2. Material and methods

## 2.1. Fly stocks and husbandry

We used females from both wild-type Dahomey and $w^{1118}$ backgrounds. Where females were mated, their partners were either Dahomey males, or $w^{1118}$ males expressing a *GFP-ProtB* construct that fluorescently labels sperm heads green [5]. All *GFP-ProtB* matings were with Dahomey females. For females that were double-mated, the second mating was to a Dahomey male into which *RFP-ProtB*, which labels sperm heads red [5], was previously backcrossed. All flies were reared under standardized larval densities (approximately 200 eggs) in bottles containing Lewis medium [18]. We collected adults as virgins under ice anaesthesia, separating them into groups of 8–12 in vials containing Lewis medium supplemented with *ad libitum* yeast granules. All flies were maintained at 25°C on a 12 : 12 light : dark cycle.

## 2.2. Experimental procedures

5-day-old virgin females from the two genotypes (Dahomey or $w^{1118}$) were individually isolated in yeasted vials under ice anaesthesia. Females were randomly allocated to mated or virgin treatments, one of three age classes for when they were to be dissected (1 day, 5 days, 9 days after the experimental matings), and, for the Dahomey females, whether their first (or only) male partner would transfer green fluorescent or non-fluorescent sperm.

24 h later we aspirated males of the relevant genotype into the mating treatment vials where they remained until the pair mated. We then transferred females into fresh, yeasted vials every 24 h for the first 3 days, and every 2 days thereafter. Additionally, on day 9, we offered 30 GFP-male-mated females the opportunity to remate with a male transferring RFP-tagged sperm to test for second male effects. 20 mated within the 4 h offered and were dissected 24 h later. All female-housing vials were retained to allow any offspring to develop and were stored at −20°C once they had enclosed ready for counting. Reproductive tracts were dissected from fresh females in PBS and sperm in the spermathecae manually counted at 40× magnification under a Motic BA210 light microscope with GFP and RFP channels. While counting, we shifted the focus to capture sperm distributed across

different focal planes. An alternative imaging set-up is required to accurately count sperm in the seminal receptacle, where a much greater number of sperm are distributed over a much greater surface area (e.g. [19,20]), so we instead recorded the presence of sperm (yes/no) in this organ. In a second experiment, we kept virgin Dahomey females in groups of 8–12 for either 7 or 26 days, flipping them onto fresh, yeasted vials every few days, to test for later life effects.

## 2.3. Statistical analysis

All analyses were performed in R (v. 3.5.1). We analysed the probability of a female exhibiting an SDP using a generalized linear model with a binomial distribution, including age (as an ordered factor), female genotype, and mated status as cofactors. We used a $\chi^2$-test to analyse differences in the proportion of 7- and 26-day-old Dahomey virgin females exhibiting SDPs. When analysing sperm numbers in fluorescent-mated females, we removed five individuals that failed to produce any offspring (5 out of 78), which is suggestive of mating failure or infertility. We analysed the number of sperm stored by females across the two spermathecae using a linear model. We used a linear mixed-effects model to analyse the number of first or second male sperm in an individual spermathecae while controlling for the identity of the female as a random effect. To analyse offspring production, we used a linear model that included male genotype (i.e. Dahomey or GFP-tagged), and the presence/absence of an SDP as factors. We analysed each female age class separately due to the bimodal distribution of offspring counts in the full dataset. In all models, we tested for the significance of factors using the 'drop1' function with a $\chi^2$-test with GLMs, or an F-test when using linear models. For linear mixed-effects models, we used Sattherthwaite's approximation to estimate degrees of freedom.

# 3. Results

## 3.1. Characterisation of SDPs

Most females had clear spermathecal ducts (e.g. figure 1*a*). However, a subset exhibited an SDP within one or both ducts (figure 1*b*–*d*). SDPs appeared similar in coloration to the spermathecae themselves. However, they showed distinct autofluorescence at wavelength 480 nm (figure 1*b*–*d*), suggesting compositional differences. SDPs appeared to form within the duct itself rather than encircling the muscular outer wall (figure 1*c*). While there often appeared to be separation from the spermathecal capsule by clear duct (figure 1*c*), SDPs occasionally continued into the capsule (figure 1*d*). In such cases, the portion of the duct that telescopes into the spermathecal capsule (the 'introvert', [8]) displayed an altered, SDP-like autofluorescence pattern (figure 1*d*). SDP size and whether it extended into the spermatheca capsule was variable between and within individuals (e.g. figure 1*d*).

## 3.2. The incidence of SDPs varies between genotypes, but is age- and mating-independent

The probability of females displaying at least one SDP was significantly higher in Dahomey compared to $w^{1118}$ females (LRT = 12.15, d.f. = 1, $p = 0.0005$; figure 2), but was unaffected by mating (LRT = 0.223, d.f. = 1, $p = 0.637$). Our data suggested a non-significant trend towards higher SDP incidence in older females, but we detected no significant interaction between age and female genotype (LRT = 4.653, d.f. = 2, $p = 0.098$) or the individual effect of age (LRT = 3.34, d.f. = 2, $p = 0.188$). Moreover, in a separate experiment, we found no significant difference in the incidence of SDPs between 7- and 26-day-old virgin Dahomey females (7-day = 4/31, 26-day = 7/31; $\chi^2 = 0.815$, $p = 0.367$). Combining $p$-values [21] from the two independent age experiments supports the lack of a significant effect of age on SDP prevalence ($p = 0.253$).

## 3.3. The presence of SDPs does not correlate with the number of sperm in the spermathecae following a single mating

The number of sperm held in individual spermathecae decreased as females aged, presumably due to use in fertilizations ($F_{2,68} = 10.47$, $p = 0.0001$; figure 3*a*), but the presence of an SDP had no significant effect (age × SDP: $F_{2,121} = 0.566$, $p = 0.569$; SDP: $F_{1,123} = 0.011$, $p = 0.916$; figure 3*a*). These results held if we analysed the combined number of sperm held across each female's two spermathecae (age × SDP: $F_{2,66} = 0.553$, $p = 0.578$; age: $F_{2,68} = 10.99$, $p < 0.0001$; SDP: $F_{1,68} = 0.36$, $p = 0.552$). We found five

bright-field    GFP

(a) Sp  Du  Sp

(b) SDPs

(c) SDP

(d) SDP  SDP  In  SDP  SDP  SR

**Figure 1.** Variable spermathecal duct morphologies. (*a*) The spermathecae of a dissected female with clear ducts. (*b*) A female with spermathecal duct presences (SDPs) in each duct. SDPs are circled in white. (*c*) The right-hand spermatheca given in (*b*) but at higher magnification. (*d*) A female showing morphologically divergent SDPs in each duct. GFP-tagged sperm are visible in some images. Sp, spermatheca; Du, spermathecal duct; SDP, spermathecal duct presence; SR, seminal receptacle; In, introvert.

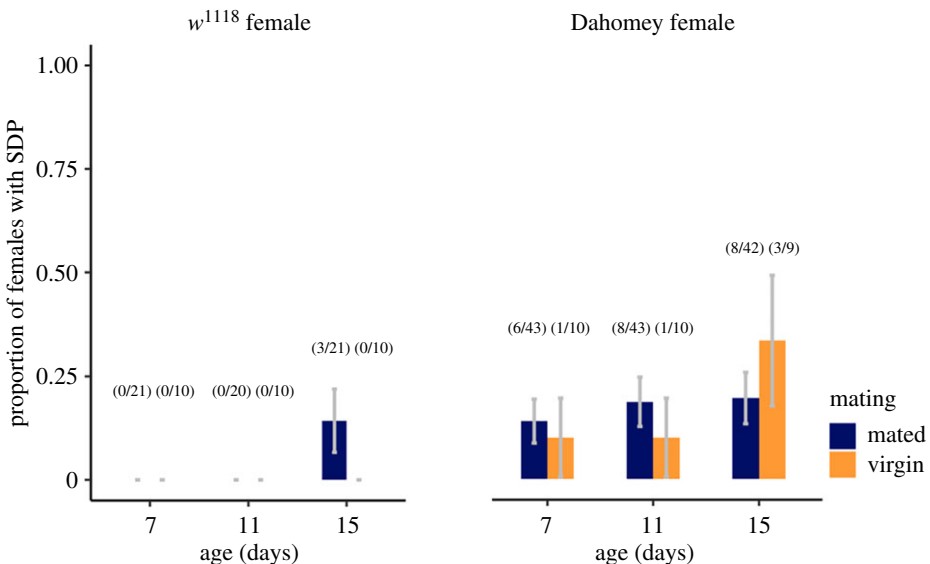

**Figure 2.** The proportion of $w^{1118}$ or Dahomey females displaying spermathecal ducts presences (SDPs) in relation to female age (in days) and mating status (mated or virgin). Mated females were mated at 6 days after eclosion. Error bars give ± 1 s.e. of the sample proportion.

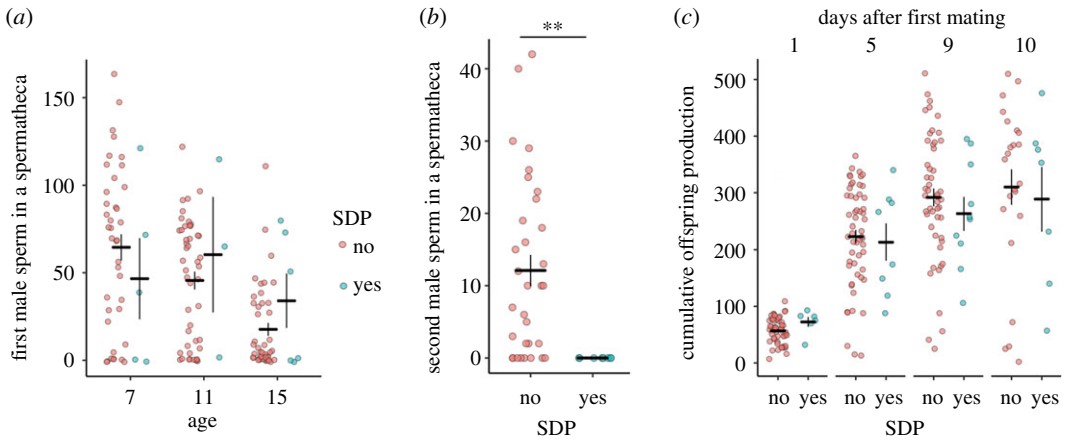

**Figure 3.** (*a*) The number of GFP-tagged sperm in a spermatheca in relation to female age and SDP presence. Females were singly mated at age 6 days. (*b*) The number of second male sperm in the spermathecae of females with (yes) and without (no) an SDP. (*c*) The cumulative number of offspring produced, plotted separately depending on whether a female had at least one SDP and in relation to the number of days after mating that the female was dissected. Both spermathecae for a single female are plotted in (*a*) and (*b*). Error bars give ± 1 s.e. of the mean. **\*\****p* < 0.01.

cases where females produced no offspring after mating (i.e. were infertile or experienced mating failure), none of which exhibited an SDP. However, this did not represent a significant increase in success for females with an SDP (Fisher's exact test, *p* = 0.583).

## 3.4. SDPs are associated with compromised storage of second male sperm

There was no significant association between SDPs and whether a female remated (proportion of rematers with SDPs = 5/20; proportion of non-rematers with SDPs = 2/10; Fisher's exact test, *p* = 1). Where females remated, the number of second male sperm was significantly lower in SDP-exhibiting spermathecae ($F_{1,31}$ = 8.824, *p* = 0.006). All 8 of the spermathecae associated with SDPs contained 0 second male sperm. This contrasted with a range of 0–42 in the 30 spermathecae (drawn from 15 females) without SDPs (figure 3*b*). The 8 SDP-containing spermathecae were drawn from five females, three of which exhibited SDPs in both ducts. Two of these females held no second male sperm in either spermathecae nor in the seminal receptacle—the only double-mating females for which this was the case.

## 3.5. SDPs do not affect the number of offspring a female produces

Combining data from all mated females, we detected no significant difference in the cumulative number of offspring produced by females with or without SDPs over any of the time points after mating (1 day: $F_{1,55}$ = 0.729, *p* = 0.397; 5 days: $F_{1,59}$ = 1.63, *p* = 0.207; 9 days: $F_{1,56}$ = 1.30, *p* = 0.260; 10 days: $F_{1,28}$ = 0.11, *p* = 0.746; figure 3*c*). The discrepancy between offspring number and the number of sperm counted in the spermathecae is due both to depletion of sperm from storage for use in fertilizations and much larger quantities of sperm being stored in the seminal receptacle compared to the spermathecae [5].

## 4. Discussion

It is unclear what SDPs are made of. Superficially, they resemble the sclerotized capsule of the spermathecae, but their distinct fluorescence pattern suggests compositional differences. Their presence in virgin females discounts a number of mechanisms through which they might form: sexually transmitted pathogens, localized immune responses to mating, or via male-derived seminal products, some of which enter into the storage organs [22–26]. The variation we detect across SDPs in terms of their size and localization may be due to genetic variation or represent different points in SDP development. SDP formation may, for example, begin within the duct before growing upwards into the introvert and the capsule itself. There is also evidence of potentially similar presences in a different strain (LHm) from a different lab (M. K. Manier, personal communication, 2020), suggesting that SDPs may be a relatively widespread and underappreciated feature of *D. melanogaster* reproductive biology.

We find no evidence that SDPs compromise the storage or release of sperm transferred by a first male, nor the number of offspring his partner ultimately produces. That said, the relatively low incidence of SDPs, coupled with the high variability generally observed in offspring and sperm counts (e.g. [19,20,27]), reduces our ability to identify any subtle effects of SDPs on reproductive performance that might exist—at least over the timescales and conditions we investigated. Although these results should be interpreted cautiously in light of the small number of individuals we find with SDPs, we do however find evidence that SDPs are associated with a failure to store sperm from a second male in the spermathecae. That we find SDPs at equal incidence in both virgin and mated females across the range of tested age classes suggests that the selective effect of SDPs on second, and not first, male sperm storage is unlikely to be due to SDPs forming between matings. The failure to detect effects on first male sperm storage and offspring production further suggests that SDPs do not function as simple plugs, restricting the passage of sperm both into and out of the spermathecae. However, it is conceivable that SDPs are modified, either by the female- or male-derived products, after the first mating in a way that restricts the entry of second male sperm into the spermathecae, but not the release of sperm that are already stored. This change could, for example, lead to SDPs disrupting mechanisms used to recruit sperm from the bursa, such as the release of chemoattractants or pressure changes that draw sperm into the spermathecal capsule [28].

An alternative explanation is that the reduced intake of second male sperm is a female age effect. A 9-day gap separated a female's first and second mating in our experiment and it is possible that the effects of SDPs on sperm storage are magnified in older females, either due to reduced tolerance or some progressive aspect to the biology of SDPs. The absence of second male sperm in both the seminal receptacle and spermathecae of some SDP-bearing females raises the possibility that SDPs are associated with wider changes to reproductive function, perhaps due to disruption to the cross-talk between seminal receptacle and spermathecae (e.g. [29]) or because SDPs are a consequence of some broader, unidentified process affecting female reproductive biology—a symptom, perhaps, of a disease state. In either case, the reduction in the storage of second male sperm may be compounded by the male if SDPs are linked to something that males can detect and discriminate against via reduced or even failure to transfer sperm.

The between-genotype differences we detect suggests standing genetic variation for SDP formation. Previous work has shown that sperm competition outcome varies with female genotype [30,31] and can be subject to male×female genotype interactions [32,33]. Variation in SDP susceptibility represents a potential mechanism through which female genotype can influence sperm competition outcome and remove the last-male sperm competition advantage observed in *Drosophila* and many other insects [34]. To explore this, future work should seek to identify the genetic contributions to SDP formation. Female reproductive tract genes already known to influence sperm competition outcome provide a useful starting point [30].

As females get older second male sperm precedence declines, but the underlying mechanism remains unresolved [35]. Our data show trends towards greater incidence of SDPs in older females, but any effect is small. It may be that SDP incidence is nonlinear with respect to age, and accelerates much later in life than we chose to study. However, given that offspring production is concentrated in the first three weeks of female post-mating life (at least in the Dahomey genetic background [36]), our data covers the ages of most reproductive relevance, and it seems likely, therefore, that age-related changes to sperm competition outcome (e.g. [35]) operate independently from SDPs.

Data accessibility. Data for the two experiments are available as electronic supplementary material.

Authors' contributions. B.R.H., I.S and S.W. designed the study and wrote the paper; B.R.H. conceived the study; B.R.H. collected and analysed data. All authors gave final approval for publication and agree to be held accountable for the work performed therein.

Competing interests. We declare we have no competing interests.

Funding. This work was funded by the EP Abraham Cephalosporin Scholarship to B.R.H with additional support from the Biotechnology and Biological Sciences Research Council (BBSRC) Doctoral Training Partnership. I.S. and S.W. were supported by a BBSRC fellowship to S.W. (grant no. BB/K014544/1) and a Dresden Senior Fellowship to S.W.

Acknowledgements. We thank Mollie Manier, Eleanor Bath, Mariana Wolfner and Yael Heifetz for thoughtful discussion, and three anonymous reviewers for their comments on earlier versions of the manuscript.

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
