## [Reviewer comments · Royal Society Open Science]

Review History

RSOS-200130.R0 (Original submission)

Review form: Reviewer 1

Is the manuscript scientifically sound in its present form?

Yes

Are the interpretations and conclusions justified by the results?

Yes

Is the language acceptable?

Yes

Do you have any ethical concerns with this paper?

No

Have you any concerns about statistical analyses in this paper?

No

Recommendation?

Accept with minor revision (please list in comments)

Comments to the Author(s)

I was reviewer 2 in a previous version of the submitted ms. I am sorry if anecdotal was the wrong choice of words. I did not mean to infer that the science done behind the study presented was not done carefully. It was meant to refer to the rare occurrence of SDPs that makes it hard to draw firm conclusions on the function or importance of these structures. An opinion I still hold on to. I still think this observation is of interest and the experiments performed are logical and well performed. But as it stands the only conclusion that can be drawn is that second male sperm is not stored. That again leaves me with more questions than answers. For example: if the formation of the structure is not mating dependent, then your suggestion that they are formed after the first, but before the second mating (lines 189-191) seems not to match that observation. The SDPs are already present in virgin females with the same frequency, so why is first sperm entry not affected and only second male sperm entry exclusively? This paradox has not been addressed by the authors. Maybe after a first mating the chemical composition/structure etc of the SDPs could change, but then only in terms of letting sperm in not letting sperm out as there is no effect on offspring production.

I also do not find the argument in line 180, as is, conclusive. Why should an impaired ability of the spermatheca to keep sperm alive, explain the absence of second sperm, but not affect the survival of first sperm as there is no difference in offspring production? First sperm have stayed longer in sperm storage organs, hence if sperm death is so acute that it kills all second sperm, what protects first sperm from these effects?

Line 185: it is not clear to me what both refers to here

Apart from these open questions I still have or that arose after the revision, I think the authors did a good job in addressing the previous comments. I also think this is an interesting observation and phenomenon, which should be made available and be shared with the scientific community.

Review form: Reviewer 2

Is the manuscript scientifically sound in its present form?

Yes

Are the interpretations and conclusions justified by the results?

Yes

Is the language acceptable?

Yes

Do you have any ethical concerns with this paper?

No

Have you any concerns about statistical analyses in this paper?

No

Recommendation?

Accept with minor revision (please list in comments)

Comments to the Author(s)

In this very well and clearly written ms, the authors describe a new structure in the *Drosophila melanogaster* female spermathecae and connect it to a blocking of second male sperm storage in these organs. They thoroughly describe its female origin and rule out that it is mating or age-dependent – all important data that rule out other potential functions and origins. I find myself

somehow agreeing with the previous referees and editor that the rare occurrence of these structures and the fact that the authors quantified them in “only” two strains, makes this finding slightly anecdotal. This opinion is validly disputed by the authors in their reply - they did perform a thorough examination that establishes this finding in the one strain in which they found them. That the field will be alerted of their existence is a worthy contribution. Whether or not they will be found in other strains and whether their function is adaptive remains to be seen and will determine the long term value of this ms. For this to happen, these data will first have to be in the public domain. This in my opinion fully justifies publication of this interesting ms.

I have a few minor comments:

1) I note an error in the authors’ reply to reviewers on page 5 starting line 8: “The sperm count data reported is just for spermathecae. Each spermathecae has a maximum capacity of ~200 (Manier et al., 2010). Larger numbers of sperm (~1000) are stored in the seminal receptacle. That partly explains the discrepancy between offspring number and sperm number. We have now added a line to the manuscript to explain this (see Lines: 158-161)”.

In fact Manier et al., 2010,(Figure 1B and C) report max 300 sperms in the seminal receptacle and max 100 in the spermathecae. So the number of sperm they report in Figure 3A is in line with the literature, but the reply to the referee is incorrect. This is OK as this does not affect the ms itself and the statement on line 158-161 makes complete sense.

2) I find that the authors did not adequately address reviewers #2’s request for clarification on how sperms were counted (Line 8 page 6 of the referee’s reports), despite indicating that they have updated that section. Would they please clarify the following three points. This would help in the replicability of this novel observation.

Line 83: Could the authors indicate how the females were frozen? Was it by dipping in liquid nitrogen or simply putting in a freezer?

Line 84: Could the authors complete that sentence by qualifying what a Motic BA210 is? I believe “light microscope” are the missing words.

Line 85: Could the authors precise how they counted sperms? For instance, was a software used? Were pictures taken at different focal points and then stitched? Please explain.

Decision letter (RSOS-200130.R0)

18-Feb-2020

Dear Dr Hopkins

On behalf of the Editors, I am pleased to inform you that your Manuscript RSOS-200130 entitled "Structural variation in *Drosophila melanogaster* spermathecal ducts and its association with sperm competition dynamics" has been accepted for publication in Royal Society Open Science subject to minor revision in accordance with the referee suggestions. Please find the referees' comments at the end of this email.

The reviewers and handling editors have recommended publication, but also suggest some minor revisions to your manuscript. Therefore, I invite you to respond to the comments and revise your manuscript.

- Ethics statement

- Data accessibility

<http://datadryad.org/submit?journalID=RSOS&manu=RSOS-200130>

- Competing interests

- Authors' contributions

- Acknowledgements

- Funding statement

Because the schedule for publication is very tight, it is a condition of publication that you submit the revised version of your manuscript before 27-Feb-2020. Please note that the revision deadline will expire at 00.00am on this date. If you do not think you will be able to meet this date please let me know immediately.

If your manuscript is newly submitted and subsequently accepted for publication, you will be asked to pay the article processing charge, unless you request a waiver and this is approved by

Royal Society Publishing. You can find out more about the charges at <https://royalsocietypublishing.org/rsos/charges>. Should you have any queries, please contact openscience@royalsociety.org.

on behalf of Prof Kevin Padian (Subject Editor)
openscience@royalsociety.org

Associate Editor Comments to Author:

While interesting questions for future research clearly remain, and the authors of this manuscript should be careful in how they respond to the queries raised by the reviewers, the manuscript may be accepted for publication if the authors take care to respond to the reviewers' comments in their revision.

Reviewer comments to Author:

Reviewer: 1

Comments to the Author(s)

I was reviewer 2 in a previous version of the submitted ms. I am sorry if anecdotal was the wrong choice of words. I did not mean to infer that the science done behind the study presented was not done carefully. It was meant to refer to the rare occurrence of SDPs that makes it hard to draw firm conclusions on the function or importance of these structures. An opinion I still hold on to. I still think this observation is of interest and the experiments performed are logical and well performed. But as it stands the only conclusion that can be drawn is that second male sperm is not stored. That again leaves me with more questions than answers. For example: if the formation of the structure is not mating dependent, then your suggestion that they are formed after the first, but before the second mating (lines 189-191) seems not to match that observation. The SDPs are already present in virgin females with the same frequency, so why is first sperm entry not affected and only second male sperm entry exclusively? This paradox has not been addressed by the authors. Maybe after a first mating the chemical composition/structure etc of the SDPs could change, but then only in terms of letting sperm in not letting sperm out as there is no effect on offspring production.

I also do not find the argument in line 180, as is, conclusive. Why should an impaired ability of the spermatheca to keep sperm alive, explain the absence of second sperm, but not affect the survival of first sperm as there is no difference in offspring production? First sperm have stayed longer in sperm storage organs, hence if sperm death is so acute that it kills all second sperm, what protects first sperm from these effects?

Line 185: it is not clear to me what both refers to here

Apart from these open questions I still have or that arose after the revision, I think the authors did a good job in addressing the previous comments. I also think this is an interesting observation and phenomenon, which should be made available and be shared with the scientific community.

Reviewer: 2

Comments to the Author(s)

In this very well and clearly written ms, the authors describe a new structure in the *Drosophila melanogaster* female spermathecae and connect it to a blocking of second male sperm storage in these organs. They thoroughly describe its female origin and rule out that it is mating or age-dependent – all important data that rule out other potential functions and origins. I find myself somehow agreeing with the previous referees and editor that the rare occurrence of these structures and the fact that the authors quantified them in “only” two strains, makes this finding slightly anecdotal. This opinion is validly disputed by the authors in their reply - they did perform a thorough examination that establishes this finding in the one strain in which they found them. That the field will be alerted of their existence is a worthy contribution. Whether or not they will be found in other strains and whether their function is adaptive remains to be seen and will determine the long term value of this ms. For this to happen, these data will first have to be in the public domain. This in my opinion fully justifies publication of this interesting ms.

I have a few minor comments:

1) I note an error in the authors’ reply to reviewers on page 5 starting line 8: “The sperm count data reported is just for spermathecae. Each spermathecae has a maximum capacity of ~200 (Manier et al., 2010). Larger numbers of sperm (~1000) are stored in the seminal receptacle. That partly explains the discrepancy between offspring number and sperm number. We have now added a line to the manuscript to explain this (see Lines: 158-161)”.

In fact Manier et al., 2010,(Figure 1B and C) report max 300 sperms in the seminal receptacle and max 100 in the spermathecae. So the number of sperm they report in Figure 3A is in line with the literature, but the reply to the referee is incorrect. This is OK as this does not affect the ms itself and the statement on line 158-161 makes complete sense.

2) I find that the authors did not adequately address reviewers #2’s request for clarification on how sperms were counted (Line 8 page 6 of the referee’s reports), despite indicating that they have updated that section. Would they please clarify the following three points. This would help in the replicability of this novel observation.

Line 83: Could the authors indicate how the females were frozen? Was it by dipping in liquid nitrogen or simply putting in a freezer?

Line 84: Could the authors complete that sentence by qualifying what a Motic BA210 is? I believe “light microscope” are the missing words.

Line 85: Could the authors precise how they counted sperms? For instance, was a software used? Were pictures taken at different focal points and then stitched? Please explain.

Author's Response to Decision Letter for (RSOS-200130.R0)

See Appendix A.

Decision letter (RSOS-200130.R1)

27-Feb-2020

Dear Dr Hopkins,

It is a pleasure to accept your manuscript entitled "Structural variation in *Drosophila melanogaster* spermathecal ducts and its association with sperm competition dynamics" in its current form for publication in Royal Society Open Science.

on behalf of Prof Kevin Padian (Subject Editor)
openscience@royalsociety.org

Appendix A

Associate Editor Comments to Author:

While interesting questions for future research clearly remain, and the authors of this manuscript should be careful in how they respond to the queries raised by the reviewers, the manuscript may be accepted for publication if the authors take care to respond to the reviewers' comments in their revision.

Reviewer comments to Author:

Reviewer: 1

Comments to the Author(s)

I was reviewer 2 in a previous version of the submitted ms. I am sorry if anecdotal was the wrong choice of words. I did not mean to infer that the science done behind the study presented was not done carefully. It was meant to refer to the rare occurrence of SDPs that makes it hard to draw firm conclusions on the function or importance of these structures. An opinion I still hold on to.

I still think this observation is of interest and the experiments performed are logical and well performed. But as it stands the only conclusion that can be drawn is that second male sperm is not stored. That again leaves me with more questions than answers. For example: if the formation of the structure is not mating dependent, then your suggestion that they are formed after the first, but before the second mating (lines 189-191) seems not to match that observation. The SDPs are already present in virgin females with the same frequency, so why is first sperm entry not affected and only second male sperm entry exclusively? This paradox has not been addressed by the authors. Maybe after a first mating the chemical composition/structure etc of the SDPs could change, but then only in terms of letting sperm in not letting sperm out as there is no effect on offspring production.

I also do not find the argument in line 180, as is, conclusive. Why should an impaired ability of the spermatheca to keep sperm alive, explain the absence of second sperm, but not affect the survival of first sperm as there is no difference in offspring production? First sperm have stayed longer in sperm storage organs, hence if sperm death is so acute that it kills all second sperm, what protects first sperm from these effects?

Line 185: it is not clear to me what both refers to here

Apart from these open questions I still have or that arose after the revision, I think the authors did a good job in addressing the previous comments. I also think this is an interesting observation and phenomenon, which should be made available and be shared with the scientific community.

- We thank the reviewer for their thoughtful comments, which exposed some flaws in our reasoning. We have now comprehensively rewritten this paragraph. See Lines 184 to 224.

Reviewer: 2

Comments to the Author(s)

*In this very well and clearly written ms, the authors describe a new structure in the *Drosophila melanogaster* female spermathecae and connect it to a blocking of second male sperm storage in these organs. They thoroughly describe its female origin and rule out that it is mating or age-dependent – all important data that rule out other potential functions and*

origins. I find myself somehow agreeing with the previous referees and editor that the rare occurrence of these structures and the fact that the authors quantified them in “only” two strains, makes this finding slightly anecdotal.

- We have since heard from another researcher who has noted similar presences in the LHm strain of *D. melanogaster*. We have included their observation as a ‘personal communication’ (see Lines 180-183).

This opinion is validly disputed by the authors in their reply - they did perform a thorough examination that establishes this finding in the one strain in which they found them. That the field will be alerted of their existence is a worthy contribution. Whether or not they will be found in other strains and whether their function is adaptive remains to be seen and will determine the long term value of this ms. For this to happen, these data will first have to be in the public domain. This in my opinion fully justifies publication of this interesting ms.

I have a few minor comments:

1) I note an error in the authors’ reply to reviewers on page 5 starting line 8: “The sperm count data reported is just for spermathecae. Each spermathecae has a maximum capacity of ~200 (Manier et al., 2010). Larger numbers of sperm (~1000) are stored in the seminal receptacle. That partly explains the discrepancy between offspring number and sperm number. We have now added a line to the manuscript to explain this (see Lines: 158-161)”.

In fact Manier et al., 2010,(Figure 1B and C) report max 300 sperms in the seminal receptacle and max 100 in the spermathecae. So the number of sperm they report in Figure 3A is in line with the literature, but the reply to the referee is incorrect. This is OK as this does not affect the ms itself and the statement on line 158-161 makes complete sense.

- We thank the reviewer for drawing our attention to that error. As the reviewer says, this error does not pertain to the ms itself.

2) I find that the authors did not adequately address reviewers #2’s request for clarification on how sperms were counted (Line 8 page 6 of the referee’s reports), despite indicating that they have updated that section. Would they please clarify the following three points. This would help in the replicability of this novel observation.

Line 83: Could the authors indicate how the females were frozen? Was it by dipping in liquid nitrogen or simply putting in a freezer?

- All females were dissected and observed fresh. The vials containing offspring were moved to a -20°C where they were stored until counting. We have clarified these procedures on lines 86 and 87.

Line 84: Could the authors complete that sentence by qualifying what a Motic BA210 is? I believe “light microscope” are the missing words.

- The reviewer is correct: this was a light microscope that incorporated a 3-channel (including RFP and GFP) fluorescence unit. We have clarified this on line 88.

Line 85: Could the authors precise how they counted sperms? For instance, was a software used? Were pictures taken at different focal points and then stitched? Please explain.

- All sperm counting was conducted manually at the microscope. We did not need to count from stitched images as there were reasonably low numbers of sperm that were distributed over a small number of focal planes. It was therefore easy to keep track of sperm across focal planes when counting (see Lines 88 to 92). This contrasts with the seminal receptacle where, in our past experience, it is almost impossible to count sperm without stitching together images taken across focal planes and tissue sections (e.g. Hopkins *et al.*, 2019a,b *PNAS*). This stitching requires a different microscopy setup, which we did not use for this study, and so we therefore refrained from counting sperm in the seminal receptacle, instead recording its binary presence/absence.